# Mutation Rate and Spectrum of the Silkworm in Normal and Temperature Stress Conditions

**DOI:** 10.3390/genes14030649

**Published:** 2023-03-04

**Authors:** Minjin Han, Jianyu Ren, Haipeng Guo, Xiaoling Tong, Hai Hu, Kunpeng Lu, Zongrui Dai, Fangyin Dai

**Affiliations:** 1State Key Laboratory of Silkworm Genome Biology, Key Laboratory of Sericultural Biology and Genetic Breeding, Ministry of Agriculture and Rural Affairs, College of Sericulture, Textile and Biomass Science, Southwest University, Chongqing 400715, China; 2WESTA College, Southwest University, Chongqing 400715, China

**Keywords:** the molecular clock, mutation rate, temperature-dependent, genome, silkworm

## Abstract

Mutation rate is a crucial parameter in evolutionary genetics. However, the mutation rate of most species as well as the extent to which the environment can alter the genome of multicellular organisms remain poorly understood. Here, we used parents–progeny sequencing to investigate the mutation rate and spectrum of the domestic silkworm (*Bombyx mori*) among normal and two temperature stress conditions (32 °C and 0 °C). The rate of single-nucleotide mutations in the normal temperature rearing condition was 0.41 × 10^−8^ (95% confidence interval, 0.33 × 10^−8^–0.49 × 10^−8^) per site per generation, which was up to 1.5-fold higher than in four previously studied insects. Moreover, the mutation rates of the silkworm under the stresses are significantly higher than in normal conditions. Furthermore, the mutation rate varies less in gene regions under normal and temperature stresses. Together, these findings expand the known diversity of the mutation rate among eukaryotes but also have implications for evolutionary analysis that assumes a constant mutation rate among species and environments.


**Significance Statement:**


Previous study asserts that the mutation rate is approximately constant among insects. Here, we found that the spontaneous mutation rate in *Bombyx mori* was significantly higher than in four previous insects, suggesting the mutation rate varies among insects. Furthermore, we found a symmetric increase in mutation rate around an optimum temperature and revealed that the mutation rate of the gene region is constant in different temperature conditions. These findings have implications for the research of evolutionary biology and the domestication history of *Bombyx mori*.

## 1. Introduction

The mutation rate (single-base substitution rate) is a central parameter of evolutionary biology [1]. Many crucial evolution inferences, such as diversity within species and divergence among species, rely on this parameter. The mutation rate of many model species has been explored, including *Escherichia coli* [2], yeast [3], *Caenorhabditis elegans* [4,5,6], *Drosophila melanogaster* [7], mouse [8], and human [9]. However, the number of species in which the rate of mutation has been studied remains limited.

The silkworm, *Bombyx mori*, is the only completely domesticated insect involved with historical trade and cultural exchange between countries of the East and West (Ancient Silk Road, ~2000 years ago), and occupies a special position among domesticated species. Thus, the detailed domestication history of *B. mori* has long been a concern to silkworm geneticists. Currently, many studies have suggested that *B. mori* was domesticated from the wild silkworm, *Bombyx mandarina*, at least 4100 years ago [10,11,12,13,14]. However, all of the domestication estimates are based on the mutation rate of *Drosophila* or *Heliconius* [15,16].

A survey of previous estimates shows that single-base substitution rates per site per generation vary significantly (~8-fold) between species. For instance, the mutation rate is 0.15 × 10^−8^ in *Caenorhabditis elegans*, 0.54 × 10^−8^ in mice, and ~1.2 × 10^−8^ in humans [4,8,9,17,18,19,20,21,22]. While previous insect studies claim that the mutation rate is approximately constant among insects [23]. However, only four insect estimates are available [7,16,23,24]. To verify the constancy of the mutation rate within insects, we studied the mutation rate of the silkworm.

Prior bacteriophage studies showed that the mutations are only selected by environmentally induced stress but are not directly stress-induced [25]. However, some studies suggest that the mutation rate can be affected by stress [26,27]. For example, bacteria studies have demonstrated that mutations are more frequent in stress-inducing environments such as those containing antibiotics [28,29,30,31]. In yeast, the mutation rate can be induced by nutrition deficit and proteotoxic stresses [3,32]. In worms and flies, the mutation rate can increase in stressful environments [6,33,34,35]. In Arabidopsis plants, elevated temperatures can increase the mutation rate [36,37]. However, the extent to which the environment can affect the genomes of animals remains unclear.

In this study, we used parents–offspring Illumina sequencing to investigate the mutation rate and spectrum of *B. mori* in normal conditions and temperature-stress-inducing environments. The results show that the mutation rate of the silkworm in normal conditions is higher than the four previously analyzed insects. Additionally, the mutation rates of the silkworm genome under stress conditions were significantly higher than those under normal conditions. Furthermore, we found that the mutation rate of gene regions of the silkworm under normal and stress conditions was constant.

## 2. Materials and Methods

### 2.1. Sample Source and DNA Extraction

The inbred domesticated silkworm (*Bombyx mori*) strain Dazao used in this study was obtained from the Silkworm Gene Bank, Southwest University, China. This strain was previously used to produce the *B. mori* reference genome [38,39]. One normal condition and two temperature-stress-inducing silkworm families were prepared (Figure 1). The recently laid eggs (aged one hour) were treated with different conditions including a normal condition (25 °C), high temperature (32 °C, 10 h), and low temperature (0 °C, 10 h). *B. mori* in larval stage is sensitive to temperature, and temperature stresses can easily lead to silkworm death. Therefore, we performed temperature stresses in the egg stage. After treatment, all individuals were reared at 25 °C under 12:12 h (L:D) photoperiod. Both parents and 30 randomly selected offspring (15 males and 15 females) in the normal family were selected to extract genomic DNA. For each stress-inducing family, DNA from both parents and 10 randomly selected offspring were extracted. The whole genomic DNA of individual moths (whole body, including all tissues) was obtained by phenol–chloroform extraction.

### 2.2. Genome Sequencing

DNA library and genome sequencing were performed by Novogene(Beijing Novogene Technology Co., Ltd., Beijing, China) For each moth, a DNA fragment library with a mean insert size of ~350 bp was constructed using the TruSeq Library Construction kit (Beijing Novogene Technology Co., Ltd., Beijing, China). Quantity and insert size of each library were checked by Qubit2.0 and Agilent 2100, respectively. Paired-end (2 × 150) sequencing reads with a mean read depth of ~30 times for each individual were obtained by Illumina HiSeq 4000 platform.

### 2.3. Identification of the Mutations

Single nucleotide variations (SNVs) were called by standard Genome Analysis Toolkit (GATK) pipeline (version 4.1.2.0) [40,41]. Reads of each individual were mapped onto the new *B. mori* reference genome (http://silkbase.ab.a.u-tokyo.ac.jp/cgi-bin/index.cgi (accessed on 12 July 2021) using BWA-MEM algorithm (Burrow-Wheeler Aligner version 0.7.15-r1140) [42]. Sam file was transformed into bam file by SAMtools (version 1.5) [43]. Picard tool (http://broadinstitute.github.io/picard/ (accessed on 15 July 2021)) was used to sort the bam file. GATK was used to mark duplicates and carry out local realignment around indels. We then called SNPs using GATK HaplotypeCaller(version 4.1.8.1). A similar criterion of the previous study was used to reduce false mutations [16]. We disregarded sites: (1) with read depth less than 10 or mapping quality less than 20, (2) marked as low quality by GATK, (3) the proportion of supported reads of mutation base less than 30%, (4) with more than 2 non-reference sites present at same read and (5) the sites of the parents that have an alternate allele or read depths less than 10. Then, the aligned reads near each mutation site were checked using an Integrated Genomics Viewer(version 2.11.9) [44]. Finally, a mutation site only present in one offspring but absent in both parents was defined as a candidate de novo mutation site (DNMs).

### 2.4. Estimation of False Positives

The de novo mutations (DNMs) were verified using PCR amplification and Sanger sequencing. For each DNM, nongenome-amplified genomic DNA of parents and mutation individuals were used as templates for PCR amplification and Sanger sequencing. We verified 105 out of 106 detected DNMs in the normal family. We could not design a proper primer paired with the excluded DNMs. We also randomly selected 20 DNMs in each stress-inducing family for verification.

### 2.5. Estimation of False Negatives

We used a previous synthetic mutation protocol to estimate false negatives [7,16]. For each family, we generated 1000 synthetic point mutations. Each synthetic point mutation was randomly selected from genomic positions and offspring. If the read depths of the selected site were *x*, we randomly selected *y* reads to change the reference base in selected points to a randomly non-reference base. The *y* value was randomly sampled from a binomial distribution with *size* = *x* and *prob* = 0.5. We then used modified reads to substitute original reads in fastq files. Finally, we realigned the changed fastq files to the reference genome and recalled SNPs using the same pipeline and criteria as the above original data.

### 2.6. Statistical Analysis and Silkworm Domestication Time Estimation

R package 4.0.2 was used to perform all statistical analyses in this study. We used previous method (compared the observed numbers of mutants versus non-mutant sites across comparisons to null expectations of discrete uniform mutation distributions, *X*^2^ test) to evaluate the significance of mutation differences among five insects [4]. To evaluate the 95% confidence interval for the mean value of the mutation rate across different treatments, genome regions, and specific bases, we used the bootstrap method with 1000 bootstrap estimates. For example, we randomly resampled 30 values from the rates of offspring of silkworms reared under normal conditions, and then the mean value of the 30 numbers was recorded as one bootstrap estimate. Finally, we sorted the 1000 bootstrap estimates from small to large and used the values at the 25th and 975th positions as the minimum and maximum values of the 95% confidence interval, respectively. Student’s *t*-test was used to evaluate the significance of mutation rate differences of the whole genome, different genomic regions, and specific bases between each stress-inducing family and normal condition family.

To estimate the domestication time of *B. mori*, the parameter estimates of population mutation parameter (*θ* = 0.034) and the time (*τ* = 0.0035) of the wild silkworm and domesticated silkworm diverged were obtained from the previous study [12]. Then, both parameters can be converted into effective population size (*Ne*) and time in years using the formulation of *Ne* = *θ*/4*µ* and in 4*Ne* units (divergence time *T* = 4*τNe*). The parameter *µ* represents the mutation rate per site per year (generally, the silkworm is raised 3 generations a year).

### 2.7. Data Access

All sequencing data were deposited to NCBI’s short read archive (project ID: PRJNA597265). The data of normal condition, low-temperature, and high-temperature treatment are accessible with BioSample accession numbers SAMN13671366-SAMN13671397, SAMN13671410-SAMN13671421, and SAMN13671422-SAMN13671433, respectively.

## 3. Results

### 3.1. Identification of the de Novo Mutations

We sequenced 56 samples (mean depth = 34.2, SD = 7.4) from three full-sib families treated with different environmental conditions including normal conditions as control, high temperature, and low temperature using parents–progeny Illumina sequencing (Figure 1). We obtained a high proportion of mapped reads (>99.6%) with an average mapping quality of 21.2 (SD = 0.2, including MQ = 0) (Appendix A). After mutation calling and filtering, we identified 106 candidate mutations from 30 offspring of the normal family. For 10 offspring from each stress treatment, we obtained 48 mutations in the family treated at high temperature and 47 mutations for the low-temperature family (Appendix A). We found that all of the mutations were heterozygous (Appendix A), suggesting that these mutations are newly generated in offspring. Further, we analyzed the distribution of the de novo mutations on the different chromosomes and found that the observed distribution was consistent with the expected distribution (assuming the mutations had a random distribution on each chromosome, *p* value > 0.2, K-S test) (Appendix A).

To estimate the true positive rate of the identified mutations, PCR amplification and Sanger sequencing were used to check the two parents and the mutation offspring. For the 106 de novo mutations identified in the normal family, we verified 84 loci except for 21 loci that gave no PCR product and for one site for which a specific primer could not be designed. We found that 79 out of 84 candidates (positive rate = ~94%) were genuine mutations (Appendix A). We also randomly selected 20 sites for verification in each stress-inducing family. The results showed that all verified 40 candidates were genuine mutations (Appendix A and Appendix A). To estimate the false negative rate, we randomly generated 1000 simulated mutation sites for each family. We then used the same protocol with the real data to call simulated mutations. We obtained 914 mutations out of 1000 sites in the normal family (false negative rate = ~8.6%), 887 sites in the high-temperature-treated family (false negative rate = ~11.3%) and 884 sites in low-temperature-treated family (false negative rate = ~11.6%).

### 3.2. The Base Mutation Bias

We found that the number of transition mutations (Ts: C↔T and G↔A) was higher than the number of transversion mutations (Tv: C↔A, C↔G, T↔A, and T↔G) in all three families. Assuming the base mutation has no bias, the ratio of Ts/Tv should be 0.5. Yet, for the three silkworm families, Ts/Tv ranged from 1.35 to 1.52 (Appendix A). In addition, the proportion of GC→AT mutation (including C:G→T:A and C:G→A:T mutations) was greater than AT→GC mutation (containing A:T→C:G and A:T→G:C) in the three silkworm families (Appendix A). We found that the GC→AT mutation bias is a common phenomenon in eukaryotic genomes by summarizing previously investigated species (Appendix A). This bias could be the reason why most eukaryotic genomes are rich in A/T bases. Further, the mutation rate of C/G bases is susceptible to temperature. There was no significant difference (*p* > 0.79, Student’s *t*-test) in the mutation rate of A/T bases among the three families (Figure 2A, Table 1, Appendix A). However, the mutation rates of C/G bases of the two temperature-stress-inducing families were significantly higher (*p* ≤ 0.05, Student’s *t*-test) than that of a normal condition family (Figure 2B, Table 1).

The number of G:C→A:T mutations was higher than the other five mutation types (including A:T→C:G, A:T→G:C, A:T→T:A, G:C→T:A, and G:C→C:G) in all three silkworm families (Figure 3A, Table 2, Appendix A). Assuming no mutation bias, the number of G:C→A:T mutations constituted about 16.7% of all de novo mutations. Nonetheless, the proportion of the G:C→A:T mutations in each silkworm family was more than 32% (Figure 3A). This tendency is a common phenomenon in eukaryotes (Appendix A). A previous study showed that the C:G→T:A transition bias could be caused by the methylation of CpG islands, which is very common in vertebrates [45]. After oxidative deamination of methylated cytosines, the cytosines are changed into thymine [46,47,48]. However, we found that the C:G→T:A transition bias in *B. mori* is not related to the methylation of cytosines at CpG islands. The C:G→T:A mutations in *B. mori* presented no CpG island distribution bias or CG dinucleotide bias (Figure 3B,C). In addition, a previous study showed that there are fewer methylated cytosines in the *B. mori* genome and that most methylcytosines occur in CG dinucleotides [49]. Therefore, the C:G→T:A transition bias in *B. mori* could be caused by a methylation-independent mechanism.

### 3.3. The Mutation Rates under Different Environments

To estimate the mutation rate of the three silkworm families, callable sites and the mutation rate of each offspring of each family were calculated (Appendix A). Then, the mutation rate was estimated to be 0.41 × 10^−8^ (95% confidence interval = 0.33 × 10^−8^–0.49 × 10^−8^, bootstrap method, see Materials and Methods) in the normal family (Table 1). This value is significantly higher (*p* = 5.1 × 10^−5^, *X^2^* test) than the mutation rate observed in four other insect species (0.28 × 10^−8^ in *Drosophila melanogaster*, 0.29 × 10^−8^ in *Heliconius melpomene*, 0.34 × 10^−8^ in *Apis mellifera* and 0.36 × 10^−8^ in *Bombus terrestris*) [7,16,23,24], suggesting that the spontaneous mutation rate of insect genomes are not as constant as previously claimed. Further, we compared the single-base mutation rate of silkworms under normal and stressed conditions and found that the mutation rate of the high-temperature family (0.58 × 10^−8^) was significantly (*p* = 0.04, Student’s *t*-test) higher than the mutation rate of the normal condition family (Figure 4A, Table 1). The mutation rate of the low-temperature family (0.57 × 10^−8^) was marginally significant (*p* = 0.06 using Student’s *t*-test; *p* = 0.04 by Wilcoxon rank-sum test) higher than that of the normal family (Figure 4A, Table 1).

To investigate the mutation rate in different genome regions, for each offspring of each silkworm family, we calculated the callable sits and the mutation rate in intergenic and gene regions (combining exons and introns and including 5′- and 3′-UTRs) (Appendix A). We found that the mutation rate of the gene region is constant (*P* = 0.76, Student’s *t*-test) in normal conditions and temperature stresses (Figure 4B). The mutation rates of the gene region in the normal condition and high-temperature- and low-temperature-treated families were 4.52 × 10^−9^ (95% CI = 3.23 × 10^−9^–5.92 × 10^−9^), 4.95 × 10^−9^ (95% CI = 2.65 × 10^−9^–7.23 × 10^−9^) and 4.92 × 10^−9^ (95% CI = 3.94 × 10^−9^–5.90 × 10^−8^), respectively (Table 1). In contrast, the rate (6.05 × 10^−9^) of the intergenic region in the high-temperature family was significantly (*p* = 0.04, Student’s *t*-test) higher than the mutation rate (3.84 × 10^−9^) of the intergenic region of the normal condition family (Figure 4C, Table 1). The mutation rate of the intergenic region in the low-temperature family (5.85 × 10^−9^) was marginally significant (*p* = 0.06 by Student’s *t*-test; *p* = 0.02 by Wilcoxon rank-sum test) higher than that of the normal condition family (Figure 4C, Table 1).

## 4. Discussion

### 4.1. The Mutation Rate Varies in Different Species

In this study, we obtained the spontaneous mutation rate (0.41 × 10^−8^) of a model insect, *Bombyx mori*. This is the fifth direct estimate of the single-nucleotide mutation rate in insects. The rate of *B. mori* is similar to the estimates in *Daphnia pulex* [50], *Canis lupus* [51], and *Ficedula albicollis* [52] and is lower than the rates of humans [18,19,20,21,22]. To the best of our knowledge, the human genome has the highest mutation rate (per site per generation) of any species for which mutation rates have been detected. That could be because the generation time of humans is much longer than other species. If we use the mutation rate per site per year, the mutation rate of silkworms (~3.74 × 10^−8^) will be much higher than that of humans (~0.048 × 10^−8^, assuming one generation time equals 25 years). In addition, the rate of *B. mori* is significantly higher (by 1.5-fold, *p*-value = 5.1 × 10^−5^, *X^2^* test) than the rate of four other insects as estimated by a parents–offspring sequencing approach [7,16,23,24] (Appendix A), which could be caused by two reasons. First, the genome size of the silkworm is higher than that of the prior four insects. We found a significant positive correlation between the mutation rate and genome size (*r* = 0.83, *p* < 0.01) among the species whose mutation rate has been estimated (Appendix A). Another possible explanation for the mutation rate differences among insects is that the generation time varies among the species. For instance, the generation time of the silkworm is ~40 days, but ~10 days for *Drosophila melanogaster*. A previous study in yeast indicated that the mutation rate per generation increases with the generation time [3]. That conclusion is partially consistent with the hypothesis of time-dependent mutagenesis [20,53,54].

Our finding suggests that prior estimates of the silkworm domestication history relying on the mutation rate of *Drosophila* or *Heliconius* requires revision. Our results suggest that the mutation rate per site per generation in the silkworm is higher (1.4~1.5-fold) than that in *Drosophila* and *Heliconius*. Based on the mutation rate estimated in this study and parameter estimates of divergence time from previously study [12], the start time of the silkworm domestication can be traced back to 9700 years ago (detailed inference see Materials and Methods). Nonetheless, we also found that the mutation rate varies in different environments (discussed below). The living environment varies significantly between wild silkworms and domestic silkworms. While the domestic silkworm lives protected, the wild silkworm lives in the wild. Is the mutation rate constant between wild silkworm and domestic silkworm? Moreover, the domesticated silkworm has a multitude of different local strains based on geographical distribution including torrid, temperate and frigid zones. Is the mutation rate constant among these local strains? The answers to these questions will help us to obtain an accurate estimate of the silkworm domestication history.

### 4.2. The Mutation Rate Varies in Different Environments

We compared the mutation rate of *B. mori* under temperature-stress-inducing environments. Compared with the normal family, the mutation rate was significantly higher in temperature-stress-inducing environments. This implies that the mutation rate can be affected by the environment, although this study does not clearly distinguish between somatic mutations and germ cell mutations. An increase in the mutation rate under stress conditions has also been observed in unicellular organisms. For instance, the mutation rate increased when bacteria experienced antibiotic and other stress treatments [28,29,30,31]. The mutation rate of yeast varies significantly under different nutrition conditions [3]. In yeast, the yearly mutation rate is less variable than the mutation rate per generation because there is a negative correlation between the yeast growth rate and the mutation rate per generation [3]. However, we did not find this trend in *B. mori*. The generation time (~40 days) did not change in the different temperature-stress-inducing silkworm families. Generally, a low temperature will lead to a longer generation time of *B. mori*, while a high temperature will speed up development and shorten the generation time [55]. However, long-time exposure to high temperatures (32 °C) or low temperatures (0 °C), especially in the larval stage, can lead to the death of *B. mori*. In contrast, the silkworm has a relatively strong tolerance to temperature in the egg stage or embryonic stage, so we observed that the generation time of *B. mori* was not affected by short high (32 °C, 10 h) and low temperature (0 °C, 10 h) treatment in the egg stage. Currently, the mechanism involving the mutation rate variation in different stresses remains unclear, which is an interesting topic for future research.

### 4.3. The Mutation Rate Is Constant in Gene Regions

Our results showed that the mutation rate in gene regions is more constant than the rate in intergenic regions. We found the mutation rate variation (differ by 1.09-fold) of gene regions among the normal and temperature-stress-inducing silkworm family was smaller than the corresponding variations (>1.5-fold) in intergenic regions. This phenomenon may be attributed to replication time-dependent mutagenesis. Generally, gene regions replicate at the early S-phase, and intergenic regions replicate at the late S-phase [56]. Single-strand DNA will appear with a higher probability at the late S-phase due to the slowing down or stalling at the replication with a decreasing dNTP pool and is more vulnerable to environmental damage [53,57,58,59]. Currently, many studies have used whole-genome sequence information to carry out evolution inference or phylogenomic analyses with sequencing technology development and reduced sequencing costs. However, we found that the mutation rate of the intergenic region that constitutes most of the genome in multicellular eukaryotes can be affected by external temporary temperatures including low temperature (0 °C) and high temperature (32 °C), which occur frequently in nature. This finding implies that using the sequence of gene regions to carry out evolution analysis assuming a constant mutation may give results that are more reliable. This finding may also help explain why there is a positive correlation between the mutation rate and genome size.

## Figures and Tables

**Figure 1 genes-14-00649-f001:**
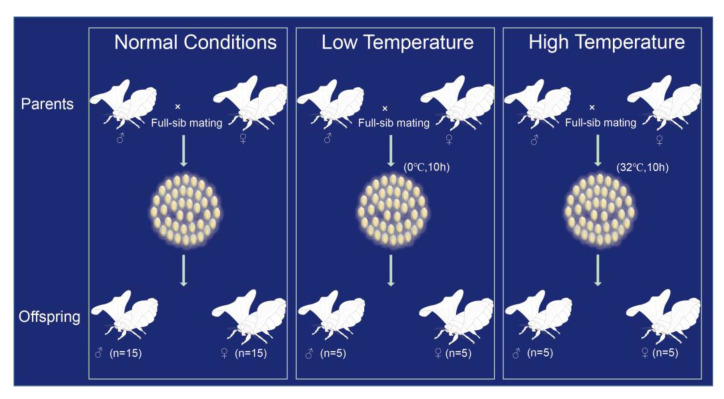
Three full-sib families were reared at different temperatures. For the normal condition family, each individual was reared at 25 °C under a 12:12 h (L:D) photoperiod. For the two temperature-stress-inducing families, eggs after one hour of oviposition were treated with either high temperature (32 °C, 10 h) or low temperature (0 °C, 10 h). Then, all treatment individuals were reared under the same conditions as the normal family.

**Figure 2 genes-14-00649-f002:**
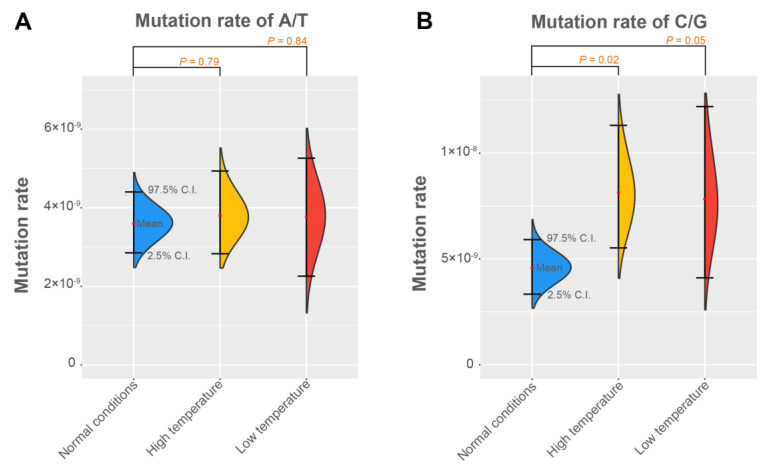
Mutation rates of A/T bases and C/G bases of the silkworm under different conditions. The mean value and 95% confidence intervals of the mutation rate of (**A**) the A/T bases and (**B**) the C/G bases. The 95% confidence interval of the mutation rate was estimated using bootstrap method with 1000 bootstrap estimates. Student’s *t*-test was used to test differences in the mutation rate (observed mutation rate per genome) between each stress-inducing family and normal condition family.

**Figure 3 genes-14-00649-f003:**
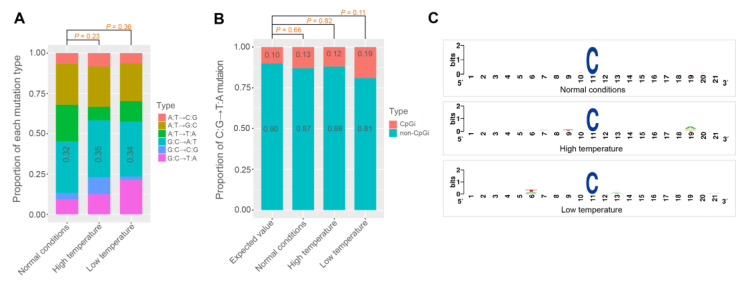
The G:C→A:T mutation bias. (**A**) The proportions of six mutation types in the silkworm under different conditions (chi-square test). (**B**) The proportions of C:G→T:A mutation sites in CpG island and non-CpG island regions. The expected value was obtained assuming the mutation follows random distribution. The total length of the CpGi region accounts for about 10% of the whole genome sequence. A two-sample test of proportions (prop.test function of R package) was used to test the significance of the difference of the distributions between observed and expected values. (**C**) The logos of flank sequences (10 bp of each side) of the C:G→T:A mutation sites in the reference genome.

**Figure 4 genes-14-00649-f004:**
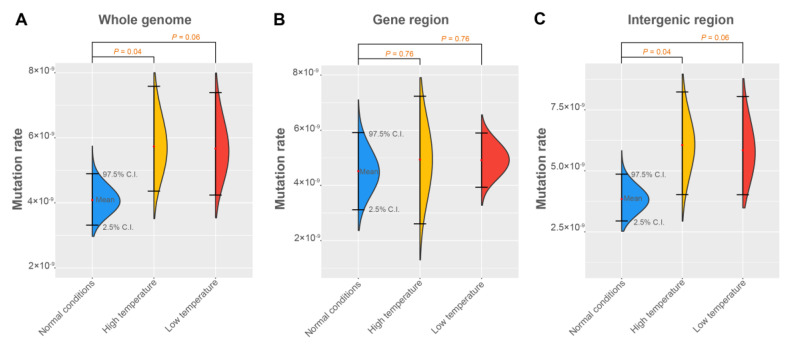
The mutation rates in whole genome, gene region, and intergenic regions of the silkworm under different conditions. The mean value and 95% confidence intervals of the mutation rate of (**A**) the whole genome, (**B**) gene region, and (**C**) intergenic region were shown. The 95% confidence interval of the mutation rate was estimated using bootstrap method with 1000 bootstrap estimates. Student’s *t*-test was used to evaluate the difference in the mutation rate between each stress-inducing family and normal condition family, and p-value is shown at the top of the figure. The data for the test of significance are the observed mutation rates (per genome) rather than the bootstrap estimates.

**Table 1 genes-14-00649-t001:** The mutation rate of different genome regions and specific bases of the silkworm under different conditions. The 95% confidence interval of the mutation rate was estimated using the bootstrap method with 1000 bootstrap estimates.

Regions and Specific Bases	Treatments	Mutation Rate (×10^−9^)		
		Mean	2.5% C.I.	97.5% C.I.
Whole genome	Normal conditions	4.09	3.31	4.90
	High temperature	5.80	4.35	7.44
	Low temperature	5.68	4.13	7.38
Gene region	Normal conditions	4.52	3.23	5.92
	High temperature	4.95	2.65	7.23
	Low temperature	4.92	3.94	5.90
Intergenic region	Normal conditions	3.84	2.94	4.81
	High temperature	6.05	4.04	8.06
	Low temperature	5.85	4.01	7.87
A/T	Normal conditions	3.60	2.91	4.34
	High temperature	3.80	3.20	4.44
	Low temperature	3.76	2.88	4.69
C/G	Normal conditions	4.58	3.34	6.01
	High temperature	8.14	6.41	9.87
	Low temperature	7.85	5.81	10.1

**Table 2 genes-14-00649-t002:** The mutation rate of different mutation types of silkworms under different conditions. The 95% confidence interval of the mutation rate was estimated using the bootstrap method with 1000 bootstrap estimates. The G:C→A:T mutation rate was highest in each treatment and is represented in bold type.

Treatments	Mutation Types	Mutation Rate (×10^−9^)		
		Mean	2.5% C.I.	97.5% C.I.
Normal conditions	A:T→C:G	0.43	0.12	0.81
	A:T→G:C	1.67	1.05	2.42
	A:T→T:A	1.49	0.99	1.98
	**G:C** **→A:T**	**3.24**	**2.19**	**4.39**
	G:C→C:G	0.38	0.00	0.86
	G:C→T:A	0.95	0.38	1.53
High temperature	A:T→C:G	0.76	0.38	1.20
	A:T→G:C	2.28	1.77	2.79
	A:T→T:A	0.76	0.44	1.08
	**G:C** **→A:T**	**4.95**	**3.59**	**6.51**
	G:C→C:G	1.45	0.77	2.13
	G:C→T:A	1.75	0.88	2.53
Low temperature	A:T→C:G	0.56	0.25	0.88
	A:T→G:C	2.07	1.44	2.75
	A:T→T:A	1.13	0.63	1.69
	**G:C** **→A:T**	**4.63**	**3.18**	**6.27**
	G:C→C:G	0.29	0.00	0.58
	G:C→T:A	2.93	1.75	4.19

## Data Availability

All sequencing data were deposited to NCBI’s short read archive (project ID: PRJNA597265). The data of normal condition, low-temperature and high-temperature treatment are accessible with BioSample accession numbers SAMN13671366-SAMN13671397, SAMN13671410-SAMN13671421 and SAMN13671422-SAMN13671433, respectively.

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
