# Peer review of "Mutation Rate and Spectrum of the Silkworm in Normal and Temperature Stress Conditions"

_genes, 2023, doi:10.3390/genes14030649_

Round 1

Reviewer 1 Report

This MS entitled “Mutation rate and spectrum of the silkworm in normal and temperature stresses conditions” by Han and colleagues describes the effects of temperature stress on genomic DNA in Bombyx mori. Mutation rate on genomic DNA is considered as indicator of evolutional speed. It is also suggested that mutation rate is influenced by the environment. The authors estimated mutation rate on B. mori genomic DNA by NGS. The effects of cold and heat stress on mutation rate also estimated by comparison between temperature stress and normal condition at the early egg stage. The data clearly showed that temperature stress enhanced mutation on genomic DNA. 

 In general, I do not see serious problems on the scientific aspects of the MS. Below are some minor points that need to be correct before the MS is accepted.

1. The authors mentioned the speed at which Bombyx mandarina evolved into Bombyx mori in introduction. But no mention about this is in discussion. Please discuss about this point in the discussion section.

2. I wondered why the authors did the heat treatment at the egg stage. Please show some explanation about this point in the MS.

3. In line 218-220, the authors wrote that they could not clarify whether genomic DNA mutation were germline mutations or somatic mutations, please state in M&M from which tissue the DNA was extracted.

4. Table 2 was split into 2 pages. Please lay out the table so that it fits on one page.

Reviewer 2 Report

This manuscript reports the mutation rates of silkworm genome under normal and stress conditions. The authors found the significantly higher mutation rates under stress conditions than normal conditions, and the mutation rate of gene regions under normal and stresses conditions was constant. Although the findings are not surprising, the large dataset of resequencing silkworm genome between parents and offspring should be of value to silkworm genome research.

Major comments.

1. Why only eggs (not worms) were treated with different conditions? The treatment duration of 10 hours seems too short to exert an effect on the spontaneous mutation. Why did not perform stress treatments during worm larva stage?  

2. How did the authors construct the background genome sequence in the study? That is, how to ensure that the heterozygous sites of the parents are all included in the genome background? This is because some heterozygous mutations of the parents are hard to identify from mapping to the reference genome.

3. Generally, the mutation rate of gene regions is lower than intergenic regions. Mutations in gene regions may cause dysfunction of the gene, thus will be largely/preferentially repaired by DNA repair enzyme. How to explain the mutation rate of gene regions (4.52) is higher than intergenic regions (3.84) in normal conditions in the manuscript?

4. The authors conclude a significant positive correlation between the mutation rate and genome size. Besides genome size, is there any specific characteristics within the genome related to the mutation rate? Such as the ratio of the sequence length of intergenic regions to gene regions within the genome, since they displayed different mutation rates.

5. Line 202-203, the rate of B. mori is lower than the rates of humans, but the generation time of humans is much longer than B. mori. Therefore, I think the authors should add the comparable analysis of approximate mutation rate with per site per year in humans in compared with per site per generation in B. mori.

6. Line 226-228,“The generation time (~40 days) did not change in the different temperature-stress-inducing silkworm families.”. If the silkworms were raised in different temperature throughout their whole lifespan, their generation time would be changed. Please discuss it.

7. Line 56-61, since the study is about temperature stresses on mutation rate and spectrum, the high temperature stress affects mutation rate of plants should be introduced, such as Arabidopsis (Genome Biology, 2021, 22: 160).

Round 2

Reviewer 2 Report

All my questions and concerns have been addressed.